# Strategic Development of Dielectric Strength Prediction Protocol for Perfluorocarbon and Nonperfluorocarbon Compounds

**Min Kyu Choi** [1] **and Ki Chul Kim** [1,2,*] 

1   Computational Materials Design Laboratory, Department of Chemical Engineering, Konkuk University, Seoul 05029, Republic of Korea
2   Division of Chemical Engineering, Konkuk University, Seoul 05029, Republic of Korea
*   Correspondence: kich2018@konkuk.ac.kr

**Abstract:** Predicting the dielectric strengths of organic compounds is critical for identifying potential insulating gases. However, experimental evaluation techniques are time-consuming, and current computational protocols are limited in scope. In this study, to develop a reliable prediction protocol for the dielectric strengths of a broad array of perfluorocarbon (PFC) and non-PFC compounds, systematic linear regression is combined with computational calculations of relevant core factors. The designed equation-based protocol is demonstrated to have four core factors, including two high-correlation factors (polarizability and molecular weight) and two critical factors (ionization energy and highest occupied molecular orbital (HOMO)–lowest unoccupied molecular orbital (LUMO) gap). The two critical factors are crucial for determining a suitable protocol, as reliable predictions of dielectric strength are only possible if the ionization energy and HOMO–LUMO gap are maintained within specified ranges for all the compounds. These findings can act as design guidelines for future computational protocols to predict the insulating properties of PFC and non-PFC compounds.

**Keywords:** dielectric strength; insulating gas; computational protocol; organic compound; regression; density functional theory

## 1. Introduction

In grid-scale energy applications, high-voltage transmission systems play a critical role in efficiently transferring energy from energy sources to destinations with minimal electricity loss [1–3]. In these systems, sulfur hexafluoride ($SF_6$), a synthetic fluorinated compound, has been widely utilized as an insulating gas to minimize electricity loss during transportation [4,5]. Advantageously, $SF_6$ is gaseous under a wide range of operating conditions owing to its low boiling point [6,7] and is extremely stable, even at temperatures higher than 500 °C [3]. As an insulating gas, $SF_6$ is beneficial in terms of its cost-effectiveness, facile manufacturing process, and recyclability [8]. Nonetheless, $SF_6$ is a greenhouse gas with a global warming potential of 23,900, indicating a capability of trapping infrared radiation much more efficiently than $CO_2$ [1,5]. Consequently, to reduce the detrimental environmental impact, $SF_6$ should be replaced with less harmful insulating gases [2,5,9,10].

Various efforts have been made to resolve this issue while maintaining a high dielectric strength similar to that of $SF_6$ [11]. A high dielectric strength is critical for insulating gases because they generally undergo electrical breakdown when a voltage exceeding the dielectric strength is applied [1,2,10,12]. Fluorinated carbon compounds, such as dichlorodifluoromethane and carbon tetrafluoride, have been shown to have high dielectric strengths compared to $N_2$ [13,14]. Furthermore, metallic nanoparticles, such as Cu and Al, have been demonstrated to weaken the insulating strength of iodomethane, increasing the electronic conductivity and decreasing the resultant electrical breakdown voltage [14]. Recently, 1,1,1,2,2,4,5,5,5-nonafluoro-4-(trifluoromethyl)-3-pentanone (also known as NOVEC 1230) was developed as a novel ecofriendly insulating gas [15].

Under an electric field of a certain strength, electrons accelerated by the electric stress apply a force to the insulating gas, which releases free electrons that subsequently collide with other insulating gas molecules, resulting in ionization and electrical conduction through kinetic energy transmission [16,17]. Consequently, the dielectric strength is regarded as a key parameter for estimating the electrical breakdown voltage of an insulating gas candidate before considering other critical parameters, such as liquefaction temperature, toxicity, chemical stability, dielectric constant, electron affinity, and byproduct phase [7,12,17,18]. With the aim of identifying promising alternative insulating gases, a few studies have focused on effectively assessing the dielectric strengths of a broad array of insulating gas candidates [10]. For instance, the insulating performance of selected insulating gas candidates was evaluated based on previously reported experimental data [19,20]. However, these experimental techniques are not suitable for rapidly evaluating the dielectric strengths of large sets of organic compounds [19,21].

In this regard, high-throughput computational approaches have been suggested as an alternative for predicting the dielectric strengths of large numbers of organic compounds [21]. For example, Zhang et al. employed a density functional theory (DFT) modeling approach combined with a linear regression technique to predict the dielectric strengths of $C_4F_7N$, $C_5F_{10}O$, and HFO-type gases [19]. To achieve reliable predictions, they constructed an equation-based function that correlated dielectric strength with polarizability and ionization energy [19,21]. This function was further developed by incorporating molecular weight as a novel parameter in the protocol and validated for a large set of perfluorocarbon (PFC) compounds, which generally have lower global warming potentials than $SF_6$ [20–22]. Despite these significant advances, computational protocols for predicting dielectric strength are still limited in their ability to identify (i) core parameters and (ii) target compounds.

In this study, the previous computational protocol, which was developed using DFT modeling combined with an equation-based function, was revisited to incorporate various parameters with the potential to accurately predict dielectric strengths. The developed computational protocol was optimized using a combination of the most suitable core parameters for not only PFC compounds but also non-PFC compounds. In particular, the strategic combination of the highest occupied molecular orbital (HOMO)–lowest unoccupied molecular orbital (LUMO) gap, which is a core electronic property, with other original parameters (i.e., polarizability, ionization energy, and molecular weight) provided the most accurate prediction of dielectric strengths for versatile compounds, which suggests a promising direction for protocol establishment.

## 2. Methodology

### 2.1. Overview of Strategy and Materials Database

In this study, the advanced computational protocol for reliably predicting the dielectric strengths of organic compounds (PFC and non-PFC compounds) comprises the following stages (Figure 1): (i) introduction of the computational protocol previously developed for predicting the dielectric strengths of PFC materials [19,21], (ii) reparameterization of the previous protocol to cover all PFC and non-PFC compounds, and (iii) full upgrade of the computational protocol through the introduction of new influencing factors. Each of these stages is described in detail in the next section. It should be noted that the previous computational protocol has been confirmed to predict the dielectric strengths of PFC compounds, but its reliability for predicting the dielectric strengths of non-PFC compounds has not yet been validated.

The materials database utilized in this study contains both PFC and non-PFC compounds (Figures 2, S1 and S2). A total of 150 PFC compounds, including 13 compounds that were not included in our previous database of 137 compounds, were designed to obtain a comprehensive PFC-based database with a broad array of fragments, such as chalcogen, halogen, and pnictogen [20]. The newly added PFC compounds are represented by chlorotrifluoromethane ($CClF_3$), hydrotrifluoromethane ($CHF_3$), and tetrafluoromethane

(CF$_4$) [5,19,22]. In addition, 18 non-PFC compounds, including N$_2$O, CO$_2$, SO$_2$, CCl$_4$, and CH$_4$, were added to the PFC-based database with the aim of finalizing the database design process. The newly added non-PFC compounds also contained a broad array of fragments [5,9,19,22]. Notably, to provide guidance on design direction, information on compounds that exhibit relatively high dielectric strengths within the 168 PFC/non-PFC compounds analyzed are listed as follows: C$_4$HClF$_8$O, C$_4$HF$_9$O, C$_4$H$_2$F$_8$O, and C$_5$F$_{10}$ with dielectric strengths higher than 3.0, exhibiting the dependence of the dielectric strength on the number of fluorine atoms. Furthermore, note that the primary objective of this study is to develop models capable of accurately predicting the dielectric strengths of both PFC and non-PFC organic compounds. Therefore, while acknowledging the significance of other properties, such as boiling point and greenhouse warming potential, in the development of insulating gas alternatives to SF$_6$, our primary focus remains on reliably predicting the dielectric strengths of the compounds.

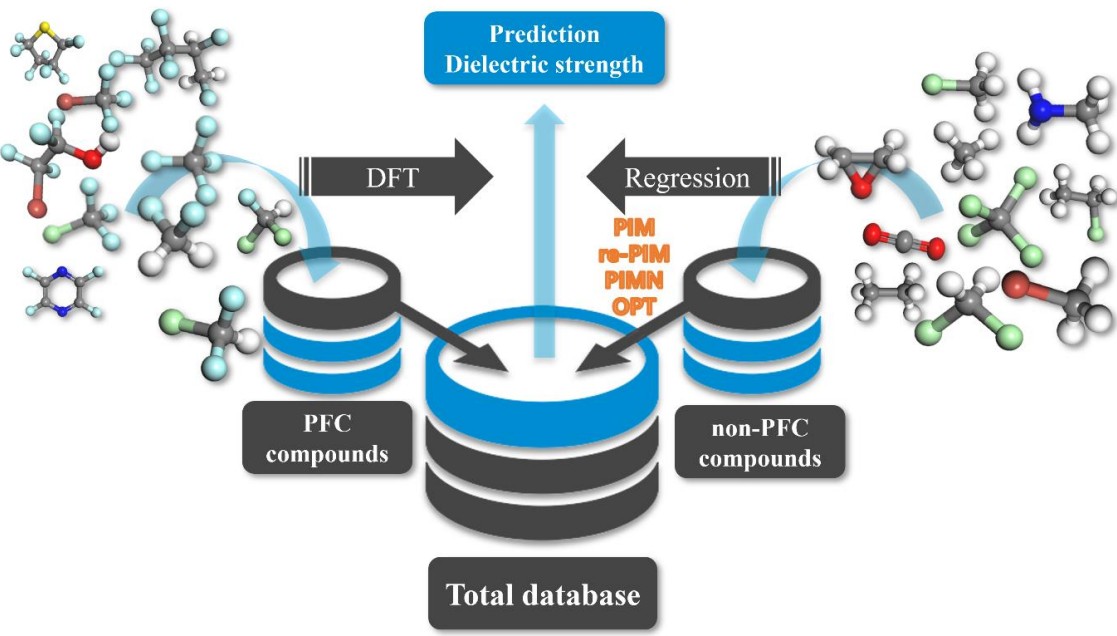

**Figure 1.** Schematic flow of the DFT-/equation-based protocol for predicting the dielectric strength of 168 PFC and non-PFC compounds.

*2.2. Computational Protocols Employed to Predict Dielectric Strength*

The dielectric strengths of the designed organic compounds were calculated using four distinct equation-based protocols: (i) POLAR-IE-MW equation (PIM), (ii) reparameterized POLAR-IE-MW equation (re-PIM), (iii) parameterized POLAR-IE-MW-NF equation (PIMN), and (iv) optimized factor equation (OPT). Here, each equation is named to indicate the incorporated factors (i.e., polarizability (POLAR), ionization energy (IE), molecular weight (MW), and new factor (NF)) with or without parameterization. In detail, the PIM ($E_{PIM}$), re-PIM ($E_{re-PIM}$), PIMN ($E_{PIMN}$), and OPT ($E_{OPT}$) equations are given as follows:

$$E_{PIM} = 0.0012\alpha^1(\varepsilon_i^a)^{0.288}(M_w)^{0.401}, \tag{1}$$

$$E_{re-PIM} = x\alpha^y(\varepsilon_i^a)^z(M_w)^m, \tag{2}$$

$$E_{PIMN} = x\alpha^y(\varepsilon_i^a)^z(M_w)^m\theta^n, \tag{3}$$

$$E_{OPT} = x\alpha^y(N_e)^z(M_w)^m\left(E_f\right)^n(V_m)^l, \tag{4}$$

where $\alpha$, $\varepsilon_i^a$, $N_e$, $M_w$, $\theta$, $E_f$, and $V_m$ are the polarizability, IE, number of electrons, MW, NF, formation energy, and molar volume, respectively. In particular, $\theta$ is the number of electrons, HOMO, HOMO–LUMO gap, electronic spatial extent gap, formation energy, or molar volume. Notably, the equations contain parameters (*x*, *y*, *z*, *m*, *n*, and *l*) to be fitted for the reliable prediction of the dielectric strength. The polarizability and IE of each compound were computed using a DFT modeling approach with the B3LYP functional and 6-311+G(d,p) basis set, as described in our previous study [21]. The DFT-computed values of these core factors have already been validated for a large set of organic compounds by systematic comparison with their experimental values [21]. Notably, the DFT modeling approach has been widely utilized for the investigation of electrochemical properties of organic compounds [23–34]. An average relative error was computed for each core factor which was predicted by the DFT modeling approach. The relative error of each compound is defined as the deviation of the DFT-calculated value from its experimental value, which is subsequently normalized by the experimental value. The average relative error is therefore the averaged value for the relative errors of 168 PFC and non-PFC compounds. In that sense, the average relative error is a descriptor to determine how accurately the DFT-based modeling approach can predict the core factors.

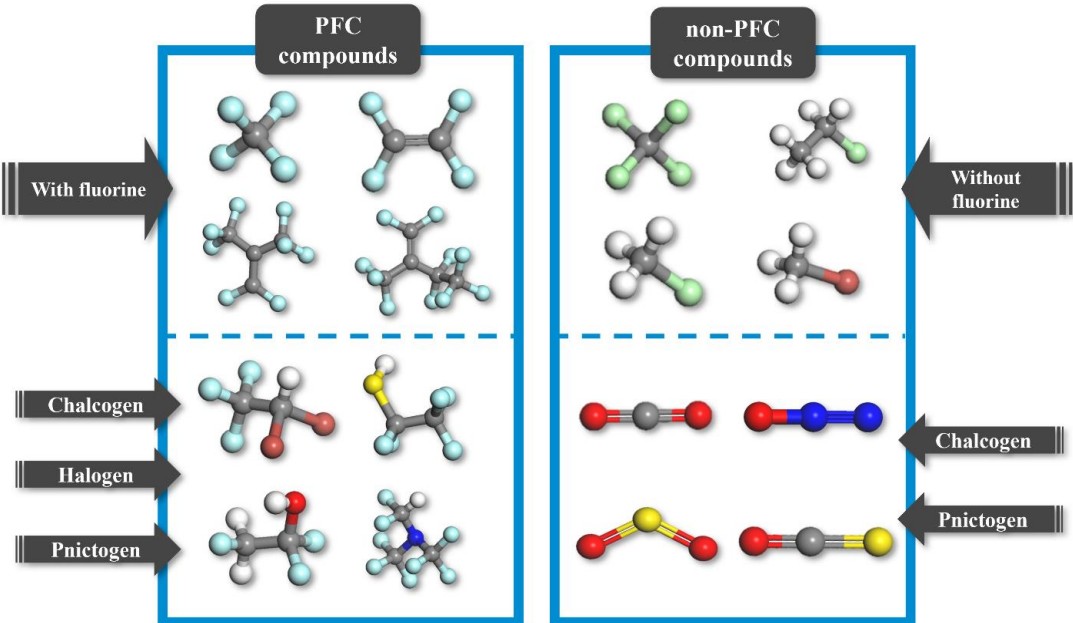

**Figure 2.** Classification of organic compounds in the database. The gray, white, blue, red, light green, brown, and sky-blue atoms correspond to carbon, hydrogen, nitrogen, oxygen, chlorine, bromine, and fluorine, respectively.

The first equation, PIM, was derived from a parameterized equation developed to accurately predict the dielectric strengths of a selected set of PFC compounds in our previous study [21]. The previous equation was developed using a limited database that did not cover the full range of available PFC compounds and contained no non-PFC compounds [21]. Consequently, PIM was reparameterized to obtain the second equation, re-PIM, with the aim of further advancing this protocol in terms of the quality of the parameters in the equation. A more advanced version of the equation, PIMN, was subsequently developed to overcome the limited combination of original factors (i.e., polarizability, IE, and MW), and was considered critically impactful for dielectric strength predictions in our previous study. This advance was achieved by incorporating a new factor, $\theta$. The new factors introduced in this study, which included electrical factors (the number of electrons, HOMO, HOMO–LUMO gap, and electronic spatial extent gap), chemical factors (formation energy), and physical factors (molar volume), are discussed in detail in the next section.

The sequential development process was completed by constructing the final equation, OPT, using a comprehensive set of desirable factors, namely polarizability, the number of electrons, MW, formation energy, and molecular volume. These factors were selected by identifying the factors that were highly correlated with the dielectric strength in the PIMN development process.

## 3. Results and Discussion

### 3.1. Universal Validity: Reliability of Previous Protocol for Predicting the Dielectric Strengths of Non-PFC Materials

The dielectric strength prediction protocol, PIM, was previously developed and validated for a selected set of PFC compounds [21]. In the current study, PIM was employed to predict the dielectric strengths of both PFC and non-PFC compounds with the aim of investigating the universal applicability of this protocol to a broad range of organic compounds (Figure 3a,b). Although PIM provided reasonable predictions for the dielectric strengths of the compounds with a trend line of $y = 0.993x + 0.035$ (Figure 3a; Table 1), the accuracy level was slightly lower than that in our previous study. Specifically, for the compounds in this study, this protocol gave a relative error of 19.83% with a root mean squared deviation (RMSD) of 0.329, and a relative error of 14.69% with an RMSD of 0.338 was obtained for the 137 PFC compounds in our previous study (Figure 3b; Table 1). Further analysis of the distributions of the core factors (i.e., polarizability, IE, and MW) constituting the PIM protocol revealed that the compounds of interest in this study cover a broader range for each factor than those in our previous study (Figure 3c). This difference may be responsible for the slight deterioration in the protocol accuracy. Furthermore, the increased structural diversity of the database containing the additional 13 PFC and 18 non-PFC compounds could also contribute the decreased accuracy.

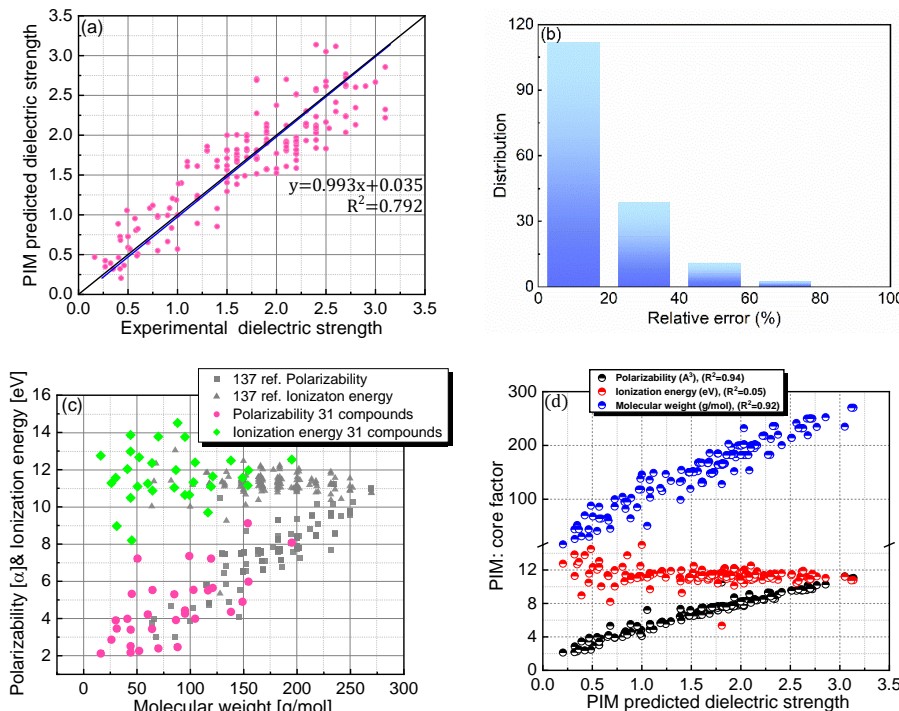

**Figure 3.** (**a**) Comparison of PIM-predicted and experimental dielectric strengths for 168 PFC and non-PFC compounds. (**b**) Distribution of relative errors for the PIM-predicted dielectric strengths. (**c**) Distributions of polarizability, ionization energy, and molecular weight for both the 137 PFC compounds in our previous study and the 31 newly added PFC/non-PFC compounds in this study. (**d**) PIM-predicted dielectric strengths as functions of the core factors (polarizability, ionization energy, and molecular weight).

**Table 1.** Developed equation-based protocols with parameterized coefficients, average relative error, and root mean squared deviation (RMSD).

| Prediction Protocol and Core Factor | | Average Relative Error | RMSD | x | y | z | m | n |
|---|---|---|---|---|---|---|---|---|
| Equation (1). PIM | | 19.83% | 0.329 | 0.0012 | 1 | 0.288 | 0.401 | - |
| Equation (2). re-PIM | | 18.94% | 0.328 | 1 | 1 | 0.327 | 0.457 | - |
| **Equation (3). PIMN** | Number of electrons ($N_e$) | 18.20% | 0.329 | 1 | 0.94 | 0.32 | 0.1 | 0.522 |
| | HOMO [eV] | 17.59% | 0.313 | 0.98 | 1.275 | 0.005 | 0.408 | 1 |
| | HOMO–LUMO gap [eV] | 16.72% | 0.289 | 1.21 | 0.916 | 0.217 | 0.732 | 0.65 |
| | Electronic spatial extent gap [a.u.] | 18.96% | 0.328 | 0.65 | 1.012 | 0.37 | 0.459 | 0.001 |
| | Formation energy [a.u.] | 18.45% | 0.320 | 0.38 | 0.83 | 0.48 | 0.461 | 0.106 |
| | Molar volume [$cm^3\ mol^{-1}$] | 17.89% | 0.346 | 1 | 0.638 | 0.281 | 0.757 | 0.253 |

| Prediction protocol and core factor | | Average relative error | RMSD | x | y | z | m | n | l |
|---|---|---|---|---|---|---|---|---|---|
| Equation (4). OPT | | 17.70% | 0.340 | 0.365 | 0.549 | 0.9 | 0.001 | 0.079 | 0.008 |

### 3.2. Strategies for Upgrading the Dielectric Strength Prediction Protocol

The PIM protocol was further developed using three distinct strategies to improve the prediction accuracy for the dielectric strengths of 168 PFC and non-PFC compounds. These development processes involved the reparameterization of PIM (strategy I), incorporation of a novel factor into PIM (strategy II), and appropriately combining highly correlated factors into an equation-based protocol (strategy III).

### 3.2.1. Strategy I: Reparameterization of Previous Protocol

For PIM, the computational protocol developed in our previous study was adopted for both the equation structure and relevant parameters (0.0012, 0.288, and 0.401). As the parameters were optimized to reliably predict the dielectric strengths of the 137 PFC compounds in the original database, they do not cover the compounds newly added in this study. Therefore, to overcome the slight deterioration in accuracy, re-PIM was developed via the reparameterization of PIM to predict the dielectric strengths of the 168 PFC and non-PFC compounds (Figure 4; Table 1). As shown in Table 1, the reparameterization produced appreciable values for all the parameters of re-PIM, implying that all the factors contribute meaningfully to the predictions of dielectric strength (Table 1). However, compared to PIM, re-PIM provides an insignificant enhancement in the accuracy level of the predictions (Figure 4a,b). Specifically, re-PIM shows a prediction ability comparable to that of PIM, with a trend line of $y = 0.978x + 0.087$ and a slightly improved accuracy (relative error = 18.94%; RMSD = 0.328). Improvements in the predication ability may be fundamentally limited by the use of the same equation-based protocol for both PIM and re-PIM.

### 3.2.2. Strategy II: Incorporation of a Novel Factor into the Previous Protocol

To overcome the limitation of the aforementioned equational framework, which is structurally fixed by the polarizability, IE, and MW, a novel factor was incorporated and parameterized (Figure 5; Table 1). The prediction accuracy of the newly designed protocol, PIMN, was expected to strongly rely on the identity of the incorporated novel factor. Three distinct types of properties, namely electronic, physical, and chemical properties, were introduced as novel factors. The number of electrons, HOMO, HOMO–LUMO gap, and electron spatial extent gap were categorized as electronic properties, whereas the formation energy and molar volume were used as chemical and physical properties, respectively. Depending on the incorporated new factor, the PIMN protocol gave relative errors of 16–18% with RMSD values of 0.289–0.346 for predicting the dielectric strengths of the 168 PFC and non-PFC compounds (Figure 5a,b). For most of the novel factors,

the accuracy level was slightly better than that of re-PIM. The highest accuracy level (relative error = 16.72%; RMSD = 0.289) was observed for the incorporation of the HOMO–LUMO gap, indicating that this novel factor is optimal for predicting the dielectric strengths of the 168 PFC and non-PFC compounds.

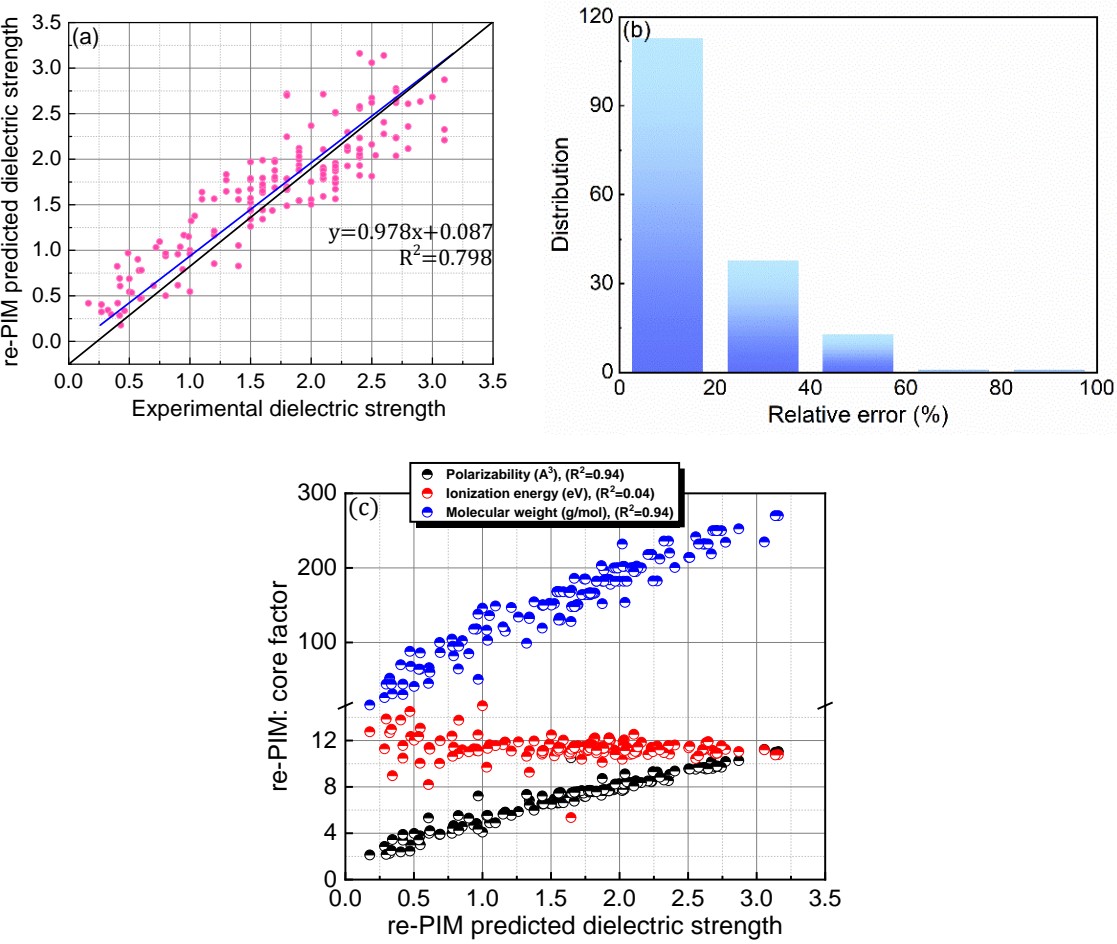

**Figure 4.** (**a**) Comparison of re-PIM-predicted and experimental dielectric strengths for 168 PFC and non-PFC compounds. (**b**) Distribution of relative errors for the re-PIM-predicted dielectric strengths. (**c**) re-PIM-predicted dielectric strengths as functions of the core factors (polarizability, ionization energy, and molecular weight).

### 3.2.3. Strategy III: Importance of Original Factors in Dielectric Strength Predictions

It is interesting to consider whether an equation-based protocol can be developed by combining factors that are highly correlated with dielectric strength (i.e., Equation (4)). Accordingly, the impact of this distinct protocol design on the prediction accuracy was explored (Figure 6; Table 1). To achieve this goal, chemical (i.e., formation energy, polarizability, ionization energy, the number of electrons, HOMO, HOMO–LUMO gap, and electronic spatial extent gap) and physical (i.e., molecular weight and molar volume) descriptors were further investigated for the 168 PFC and non-PFC compounds (Figure S3). The chemical descriptors were computed by the DFT method. For instance, the formation energy, which is defined as the energy required for the formation of a compound from its elemental constituents, was predicted by the DFT-calculated energies of the compound and its elemental constituents. The DFT-calculated frontier orbital analyses were employed for HOMO and LUMO values, while the DFT computation of the dipole-based electric field was used for polarizability. The DFT-calculated energy change associated with oxidation was utilized for ionization energy. The electronic spatial extent gap is a measure of how widely the electronic density is distributed within a molecule. It reveals the av-

erage distance between the electrons and the nucleus in a molecule, providing insight into its electronic structure and properties. The electronic spatial extent gap of a molecule was computed by $\int \varphi(r)^2 r^2 dv$, where $\varphi(r)$ represents the electron wave function of the molecule, $r$ is the distance between an electron and its molecular center, and $dv$ is a 3D element of space, in other words, the integration of $\varphi(r)$ and $r^2$ over three-dimensional space. Direct correlations of these descriptors with experimental dielectric strength were further analyzed for the 168 PFC and non-PFC compounds (Figure S3). As seen in the figures, the correlation degree depends on the introduced descriptor. Consequently, five distinct factors (polarizability, the number of electrons, MW, formation energy, and molar volume) were chosen as the most highly ranked factors in terms of correlation with dielectric strength (Figure S3). Notably, the IE, which is a main contributor in the original PIM protocol, as well as crucial core factors, such as the HOMO–LUMO gap and HOMO, were excluded from this equation-based protocol in PIMN. Unexpectedly, the parameterization of the developed equation-based protocol resulted in a deterioration in prediction accuracy (relative error = 17.70%; RMSD = 0.340) compared with PIMN (*via HOMO–LUMO gap* and *HOMO*) (Figure 7a,b). This result implies that the synergistic effect of appropriately selected core factors, such as polarizability, IE, MW, and HOMO–LUMO gap, is more impactful for predicting dielectric strength than the correlating ability of an individual factor.

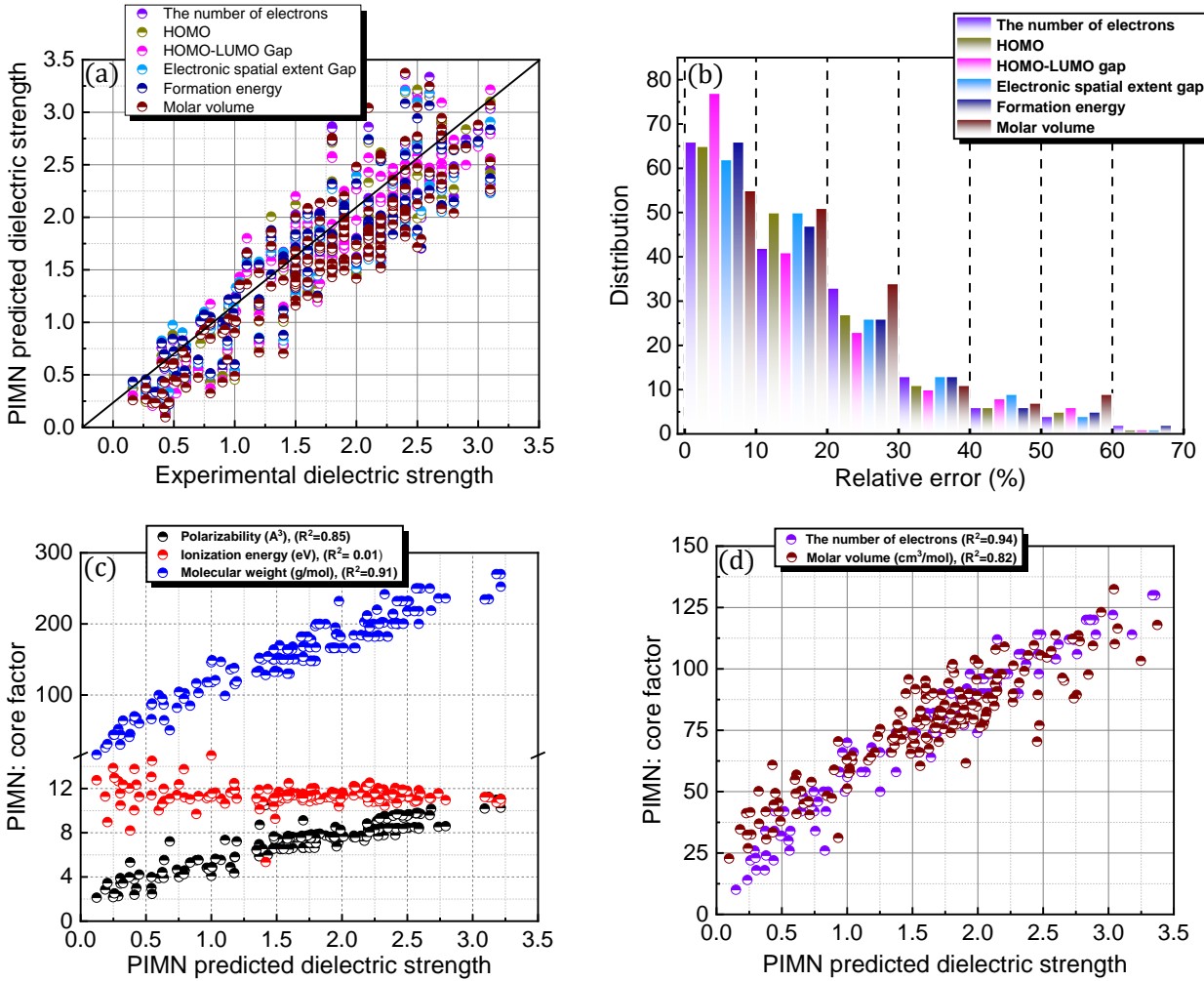

**Figure 5.** *Cont.*

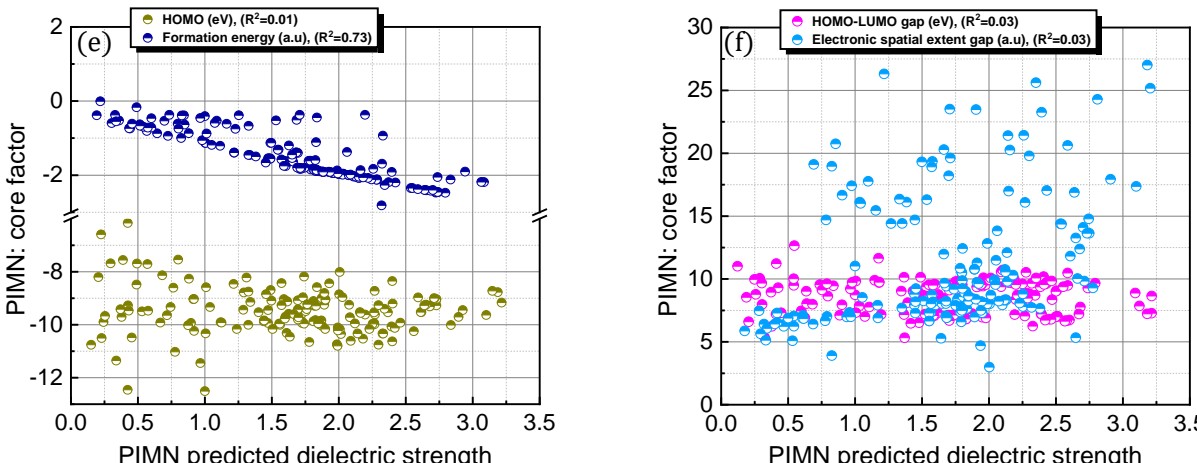

**Figure 5.** (**a**) Comparison of PIMN-predicted (based on five distinct novel factors) and experimental dielectric strengths for 168 PFC and non-PFC compounds. (**b**) Distribution of relative errors for the PIMN-predicted dielectric strengths. In (**b**), the distribution for each core factor has blocks of 0–10, 10–20, 20–30, 30–40, 40–50, 50–60, and 60–70. (**c**–**f**) PIMN-predicted dielectric strength as functions of the core factors (polarizability, ionization energy, molecular weight, and novel factors).

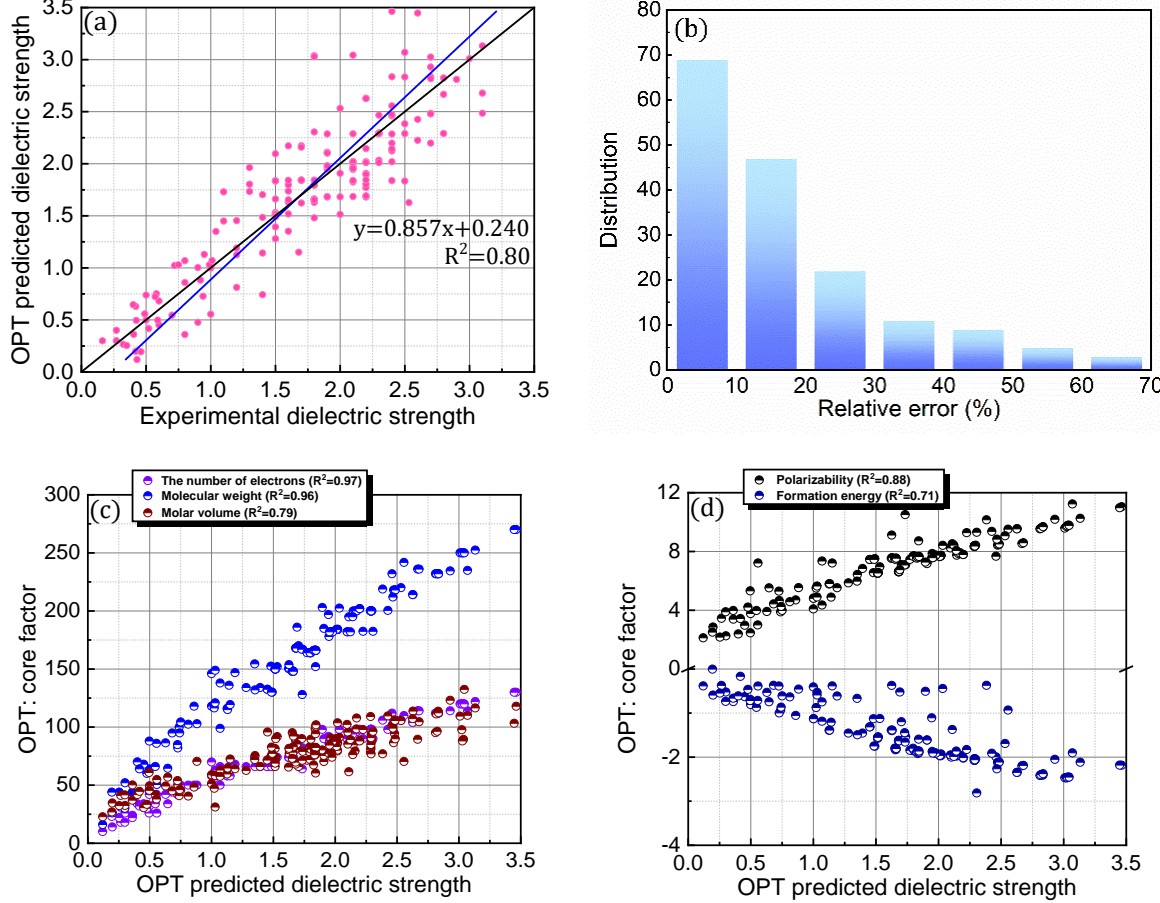

**Figure 6.** (**a**) Comparison of OPT-predicted and experimental dielectric strengths for 168 PFC and non-PFC compounds. (**b**) Distribution of relative errors for OPT-predicted dielectric strengths. (**c**,**d**) OPT-predicted dielectric strengths described as functions of the core factors (number of electrons, molecular weight, molar volume, polarizability, and formation energy).

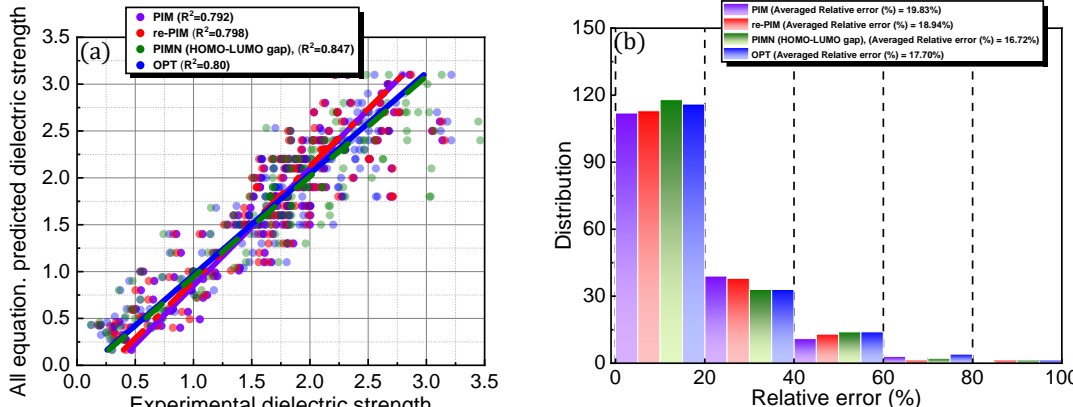

**Figure 7.** (**a**) Comparison of PIM-, re-PIM-, PIMN-, and OPT-predicted dielectric strengths with experimental dielectric strengths and (**b**) distributions of their relative errors. In (**b**), the distribution for each prediction protocol has blocks of 0–10, 10–20, 20–30, 30–40, 40–50, 50–60, and 60–70.

### 3.3. In-Depth Analysis of Correlations between Dielectric Strength and Various Core Factors

Notably, the above-discussed equation-based prediction protocols have various core factors, (i.e., polarizability, IE, MW, and novel factors) in their equational forms. Consequently, one may wonder how strongly the core factors are correlated with the predicted dielectric strength. Furthermore, to obtain a deeper understanding of the developed prediction protocols, it should be determined whether higher accuracy is achieved by incorporating core factors with higher correlation degrees. Therefore, the correlation degrees between the core factors and the predicted dielectric strengths of the 168 PFC and non-PFC compounds were examined for each equation-based protocol to associate the correlation degree with the prediction accuracy.

First, the original core factors (polarizability, IE, and MW) were correlated with the predicted dielectric strengths of the 168 PFC and non-PFC compounds (Figures 3d, 4c and 5c; Table 2). Two of these core factors (i.e., polarizability and MW) exhibited linear correlations with the PIM-/re-PIM-/PIMN-predicted dielectric strengths ($R^2 > 0.8$). It should be noted that the $R^2$ value describes the degree of deviation of a data set from a linear regression line. Thus, a compound with a higher polarizability or IE would have a higher dielectric strength. In contrast, the IE was maintained within a range of 8–14 eV for all the compounds, indicating that the dielectric strength is insensitive to this core factor. *This observation also highlights that the IE should be within the aforementioned range regardless of the type of compound, which is a key condition for achieving reliable dielectric strength predications.* This may account for the unexpected decrease in the accuracy of the dielectric strength prediction with the OPT, which did not include the IE as a core factor, as compared to PIMN (*via HOMO–LUMO gap*).

Novel core factors introduced in the PIMN and OPT protocols were further correlated with the predicted dielectric strengths of the 168 PFC and non-PFC compounds (Figures 5d–f and 6c,d; Table 2). Notably, the predicted dielectric strengths exhibited significant correlations with the number of electrons, molar volume, and formation energy ($R^2 > 0.7$). In particular, among the novel factors, the number of electrons had the strongest correlation with the dielectric strength ($R^2 = 0.94$), indicating its high contribution to determining the dielectric strength of a compound. In contrast, the HOMO–LUMO gap had a weak correlation with the dielectric strength ($R^2 = 0.03$), implying a negligible contribution to determining the dielectric strength. Importantly, as discussed for the IE, the reliable prediction of dielectric strength strongly relies on the HOMO–LUMO gap, which should be maintained within the range of 5–13 eV for all the compounds. Consequently, constraints on the values of these critical factors (IE and HOMO–LUMO gap) should be incorporated into the equation-based protocol.

**Table 2.** Linear regression lines and correlation coefficients ($R^2$) of the correlation between the dielectric strength and each core factor for each prediction protocol.

| Prediction Protocol | Core Factor | Linear Regression Line | Correlation Coefficient ($R^2$) | Corresponding Figure |
|---|---|---|---|---|
| Equation (1). PIM | Polarizability [Å$^3$]<br>Ionization energy [eV]<br>Molecular weight [g mol$^{-1}$] | $y = 3x + 2.037$<br>$y = -0.312x + 11.875$<br>$y = 82.24x + 21.111$ | 0.94<br>0.05<br>0.92 | Figure 3d |
| Equation (2). re-PIM | Polarizability [Å$^3$]<br>Ionization energy [eV]<br>Molecular weight [g mol$^{-1}$] | $y = 2.24x + 2.924$<br>$y = -0.280x + 11.814$<br>$y = 80.99x + 25.267$ | 0.94<br>0.04<br>0.94 | Figure 4c |
| Equation (3). PIMN | Polarizability [Å$^3$]<br>Ionization energy [eV]<br>Molecular weight [g mol$^{-1}$] | $y = 2.53x + 2.822$<br>$y = -0.167x + 11.629$<br>$y = 73.18x + 36.187$ | 0.85<br>0.01<br>0.91 | Figure 5c |
| | Number of electrons<br>Molar volume [cm$^3$ mol$^{-1}$] | $y = 35.97x + 17.230$<br>$y = 27.509x + 32.502$ | 0.94<br>0.82 | Figure 5d |
| | HOMO [eV]<br>Formation energy [a.u.] | $y = -0.12x - 9.312$<br>$y = -0.827x - 0.137$ | 0.01<br>0.73 | Figure 5e |
| | HOMO–LUMO gap [eV]<br>Electronic spatial extent gap [a.u.] | $y = -0.103x + 8.779$<br>$y = 2.708x + 6.578$ | 0.03<br>0.03 | Figure 5f |
| Equation (4). OPT | Number of electrons<br>Molecular weight [g mol$^{-1}$]<br>Molar volume [cm$^3$ mol$^{-1}$] | $y = 34.459x + 18.84$<br>$y = 72.137x + 36.101$<br>$y = 25.90x + 32.641$ | 0.97<br>0.96<br>0.79 | Figure 6c |
| | Polarizability [Å$^3$]<br>Formation energy [a.u.] | $y = 2.486x + 2.836$<br>$y = -0.712x - 0.301$ | 0.88<br>0.71 | Figure 6d |

## 4. Conclusions

In this study, a systematic regression method was combined with DFT-based calculations of core factors to develop a reliable equation-based protocol for accurately predicting the dielectric strengths of a broad array of PFC and non-PFC compounds. Four distinct protocols were sequentially designed through a series of stepwise upgrades of a previously developed equation-based protocol for predicting the dielectric strengths of a selected set of PFC compounds. This investigation enabled us to identify an optimal combination of core factors for designing a prediction protocol. The most accurate protocol not only included core factors that were highly correlated with the dielectric strength (i.e., polarizability and MW) but also other core factors (i.e., IE and HOMO–LUMO gap), suggesting that a critical condition exists for the reliable prediction of dielectric strength. *Notably, this study enabled the identification of this critical condition, namely, the IE and HOMO–LUMO gap should be maintained within specified ranges for all the compounds.* The findings of this study will provide useful guidance for the design of a suitable protocol to predict the insulating properties of PFC and non-PFC compounds.

**Supplementary Materials:** The following supporting information can be downloaded at: https://www.mdpi.com/article/10.3390/app13074318/s1. Figure S1. A total of 150 PFC compounds introduced in this study. Atoms with gray, white, red, blue, cyan, light green, and yellow in color depict carbon, hydrogen, oxygen, nitrogen, fluorine, chlorine, and sulfur, respectively. Figure S2. A total of 18 non-PFC compounds introduced in this study. Atoms with gray, white, red, blue, brown, light green, and yellow in color depict carbon, hydrogen, oxygen, nitrogen, bromine, chlorine, and sulfur, respectively. Figure S3. Correlations of experimental dielectric strength with (a) DFT-computed polarizability, (b) DFT-computed ionization energy, (c) the number of electrons, (d) DFT-computed HOMO, (e) DFT-computed HOMO-LUMO gap, (f) DFT-computed electronic spatial extent gap, (g) DFT-computed formation energy, (h) molecular weight, and (i) DFT-computed molar volume. Table S1. Chemical formula and IUPAC name for each of 150 PFC compounds drawn in Figure S1. Table S2. Chemical formula and IUPAC name for each of 18 non-PFC compounds drawn in Figure S2.

**Author Contributions:** Writing—review and editing, and supervision, K.C.K.; validation, formal analysis, investigation, and data curation, M.K.C. All authors have read and agreed to the published version of the manuscript.

**Funding:** This work was also supported by the Korea Electric Power Corporation (Grant number: R21XO01-4).

**Informed Consent Statement:** Not applicable.

**Data Availability Statement:** No data is shared by the authors.

**Conflicts of Interest:** The authors declare no conflict of interest.

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
