# Peer review of "Strategic Development of Dielectric Strength Prediction Protocol for Perfluorocarbon and Nonperfluorocarbon Compounds"

_applsci, doi:10.3390/app13074318_

Round 1

Reviewer 1 Report

There exist some grammar and wording problems. Please double check and revise. 

Author Response

Response: Thank the reviewer for the careful review. As suggested by the reviewer, we have corrected grammar errors through the review of the manuscript.

Reviewer 2 Report

   This paper aims at designing QSPR models that enable the dielectric strength of gaseous perfluorocarbons (PFC) and non-PFC systems to be predicted. The search for such computational tools is justifed by the fact that SF6 commonly used as an insulating material is simultaneously a greenhouse gas.

    The concept presented in this work is not new. QSPR models predicting dielectric constants have already been published. In the current work, the authors validated the published models for a larger set of insulators, reparametrized them, as well as extended the existing models by adding new descriptors like the number of electrons, HOMO-LUMO gap, etc.

Major:

1)     How are the average relative errors showed in Tabke 1 for the number of electrons, HOMO, HOMO-LUMO gap, electronic spatial extent gap, formation energy, and molar volume calculated? I understand that the average error of PIM (E_PIM), for instance, is calculated as relative standard deviation, where the deviations are calculated with regard to the experimental values of dielectric strength. But how the average errors of molecular descriptors are calculated and what is their meaning seems to be more than unclear.

2)     What is the meaning of “formation energy” and “electronic spatial extent gap”. How have they been calculated?

3)     Although the QSPR models presented in this work seem to have better predictive power than the published ones, I do not see any proposal of a strategy enabling a rational design of new insulating gases.

4)     How would the authors like to test the greenhouse properties of the new insulator? Without such a tool it would be possible to propose an “ideal” insulator that, however, has greenhouse gas properties stronger than SF6 has.

Minor:

1)     The format used on page 5 is different from that employed for the rest of the paper. It does not look good and leaves an impression that the article has been written in a hurry.

2)     Something is wrong with the format of Table 1. I do not understand the current position of the „Equation 3. PIMN” symbol.

Author Response

Reviewer 2

Comments and Suggestions for Authors

This paper aims at designing QSPR models that enable the dielectric strength of gaseous perfluorocarbons (PFC) and non-PFC systems to be predicted. The search for such computational tools is justified by the fact that SF6 commonly used as an insulating material is simultaneously a greenhouse gas.

The concept presented in this work is not new. QSPR models predicting dielectric constants have already been published. In the current work, the authors validated the published models for a larger set of insulators, reparametrized them, as well as extended the existing models by adding new descriptors like the number of electrons, HOMO-LUMO gap, etc.

Response: We appreciate the reviewer for the encouraging comments. We believe that the manuscript has been significantly improved through reflecting the comments and suggestions provided by the reviewer.

Major:

1) How are the average relative errors showed in Table 1 for the number of electrons, HOMO, HOMO-LUMO gap, electronic spatial extent gap, formation energy, and molar volume calculated? I understand that the average error of PIM (E_PIM), for instance, is calculated as relative standard deviation, where the deviations are calculated with regard to the experimental values of dielectric strength. But how the average errors of molecular descriptors are calculated and what is their meaning seems to be more than unclear.

Response: The reviewer raised an important question on the detailed information of the average relative errors. As expected by the reviewer, the relative error of each compound is defined as the deviation of the DFT-calculated value from its experimental value, which is subsequently normalized by the experimental value. The average relative error is therefore the averaged value for the relative errors of 168 PFC and non-PFC compounds. In that sense, the average relative error is a descriptor to determine how accurately the DFT-based modeling approach can predict the core parameters.

As suggested by the reviewer, we have included detailed information of the average relative error in the manuscript.

Page 4: “An average relative error was computed for each core factor which was predicted by the DFT modeling approach. The relative error of each compound is defined as the deviation of the DFT-calculated value from its experimental value, which is subsequently normalized by the experimental value. The average relative error is therefore the averaged value for the relative errors of 168 PFC and non-PFC compounds. In that sense, the average relative error is a descriptor to determine how accurately the DFT-based modeling approach can predict the core factors.”

2) What is the meaning of “formation energy” and “electronic spatial extent gap”. How have they been calculated?

Response: Thank the reviewer for the question on the definition of core factors, namely formation energy and electronic spatial extent gap, and their approaches.

The formation energy, which is defined as the energy required for the formation of a compound from its elemental constituents, was predicted by the DFT-calculated energies of the compound and its elemental constituents. In other words, the formation energy is simply the difference in the DFT-calculated energy between the compound and its elemental constituents.

Electronic spatial extent gap is a measure of how widely the electronic density is distributed within a molecule. it reveals the average distance between the electrons and the nucleus in a molecule, providing insight into its electronic structure and properties. Electronic spatial extent gap of a molecule was computed by , where  represents the electron wave function of the molecule,  is the distance between an electron and its molecular center, and  is a 3D element of space, meaning the integration of  and  over three-dimensional space.

As suggested by the reviewer, we have included detailed information of the DFT-calculated factors, formation energy and electronic spatial extent gap, in the manuscript.

Page 9: “The formation energy, which is defined as the energy required for the formation of a compound from its elemental constituents, was predicted by the DFT-calculated energies of the compound and its elemental constituents.”

Page 9: “Electronic spatial extent gap is a measure of how widely the electronic density is distributed within a molecule. it reveals the average distance between the electrons and the nucleus in a molecule, providing insight into its electronic structure and properties. Electronic spatial extent gap of a molecule was computed by , where  represents the electron wave function of the molecule,  is the distance between an electron and its molecular center, and  is a 3D element of space, meaning the integration of  and  over three-dimensional space.”

3) Although the QSPR models presented in this work seem to have better predictive power than the published ones, I do not see any proposal of a strategy enabling a rational design of new insulating gases.

Response: We appreciate the reviewer for the comment on the possibility of this study for suggesting a desired design direction of novel insulating gas candidates.

We agree with the reviewer that this study does not explore a rational design direction to developing promising insulating gas candidates. However, please note that the primary objective of this study was to develop models capable of accurately predicting the dielectric strengths of both PFC and non-PFC organic compounds. As such, our analysis focused on examining 168 PFC/non-PFC compounds that had been previously studied and their reported dielectric strength with the aim of reproducing their values.

Thus, it is not within the scope of this study to explore a rational design direction for the development of novel compounds with high dielectric strengths. Nonetheless, to provide guidance on design direction, we have included information on compounds that exhibit relatively high dielectric strengths within the 168 PFC/non-PFC compounds analyzed.

Page 3: “Notably, to provide guidance on design direction, information on compounds, that exhibit relatively high dielectric strengths within the 168 PFC/non-PFC compounds analyzed, are listed as follows: C4HClF8O, C4HF9O, C4H2F8O, and C5F10 with dielectric strengths higher than 3.0, exhibit-ing the dependence of the dielectric strength on the number of fluorine atoms.”

4) How would the authors like to test the greenhouse properties of the new insulator? Without such a tool it would be possible to propose an “ideal” insulator that, however, has greenhouse gas properties stronger than SF6 has.

Response: The reviewer raised a question on how to test the greenhouse effects of the insulating gases.

We agree with the reviewer that the greenhouse properties of insulating gas candidates, such as greenhouse warming potential, would be essentially needed to assess their potential for replacing SF6. However, please note that the primary objective of this study was to develop models capable of accurately predicting the dielectric strengths of both PFC and non-PFC organic compounds. Thus, it is not within the scope of this study to explore properties other than dielectric strength. As such, we focused on the dielectric strengths of the compounds rather than their greenhouse properties.

A relevant statement has been included in the manuscript to clarify the primary objective of this study.

Page 3: “Further, note that the primary objective of this study is to develop models capable of accurately predicting the dielectric strengths of both PFC and non-PFC organic compounds. Therefore, while acknowledging the significance of other properties, such as boiling point and greenhouse warming potential, in the development of insulating gas alternatives to SF6, our primary focus remains on reliably predicting the dielectric strengths of the compounds.”

Minor:

1) The format used on page 5 is different from that employed for the rest of the paper. It does not look good and leaves an impression that the article has been written in a hurry.

Response: Thank the reviewer for the careful review. As suggested by the reviewer, we have corrected the format on page 5 for consistency.

2) Something is wrong with the format of Table 1. I do not understand the current position of the „Equation 3. PIMN” symbol.

Response: Thank the reviewer for the careful review. As suggested by the reviewer, we have corrected the format of Table 1 for clarity.

Reviewer 3 Report

The work under review is devoted to the important and topical topic of predicting the properties of materials based on modern computational protocols that are relatively cheap compared to experiment. Actually, this is the main problem of computational quantum chemistry. In this compound, a prediction is made of the dielectric strength of organic compounds for the possibility of using them as potential insulating gases. The developed protocol based on the equation is shown to have four main factors, including two highly correlated factors (polarizability and molecular weight) and two critical factors (ionization energy and the gap between the highest occupied molecular orbital (HOMO) and the lowest unoccupied molecular orbital (LUMO)). To estimate the two critical factors, the ionization energy and the HOMO-LUMO gap, DFT simulations with the B3LYP functional and the 6-311+G(d,p) basis set are used. The authors claim to have compared the obtained values for a large set of organic compounds by systematic comparison with their experimental values. The functionality is well known and tested many times. The basis is standard, but probably sufficient for such molecules.

The work is well illustrated. Pictures correspond to the text. Interesting information with direct DFT calculations is placed in the appendix. It seemed to me that it needed a more detailed description, either in the text or in the accompanying materials themselves. Minor remarks, for example, "why is there a dotted line in Fig. 6 b)?" are of no fundamental importance. In general, I believe that the peer-reviewed work can be published as it is.

Author Response

Reviewer 3

Comments and Suggestions for Authors

The work under review is devoted to the important and topical topic of predicting the properties of materials based on modern computational protocols that are relatively cheap compared to experiment. Actually, this is the main problem of computational quantum chemistry. In this compound, a prediction is made of the dielectric strength of organic compounds for the possibility of using them as potential insulating gases. The developed protocol based on the equation is shown to have four main factors, including two highly correlated factors (polarizability and molecular weight) and two critical factors (ionization energy and the gap between the highest occupied molecular orbital (HOMO) and the lowest unoccupied molecular orbital (LUMO)). To estimate the two critical factors, the ionization energy and the HOMO-LUMO gap, DFT simulations with the B3LYP functional and the 6-311+G(d,p) basis set are used. The authors claim to have compared the obtained values for a large set of organic compounds by systematic comparison with their experimental values. The functionality is well known and tested many times. The basis is standard, but probably sufficient for such molecules.

The work is well illustrated. Pictures correspond to the text. Interesting information with direct DFT calculations is placed in the appendix. It seemed to me that it needed a more detailed description, either in the text or in the accompanying materials themselves. Minor remarks, for example, "why is there a dotted line in Fig. 6 b)?" are of no fundamental importance. In general, I believe that the peer-reviewed work can be published as it is.

Response: We appreciate the reviewer for the positive comments on the manuscript with valuable suggestion on a detailed description of DFT-calculated information, such as polarizability and ionization energy, a minor question on a dotted line in Figure 6b.

As suggested by the reviewer, we have included detailed information of the DFT-calculated factors, such as polarizability and ionization energy, in the manuscript.

Page 9: “To achieve this goal, chemical (i.e., formation energy, polarizability, ionization energy, the number of electrons, HOMO, HOMO-LUMO gap, and electronic spatial extent gap) and physical (i.e., molecular weight and molar volume) descriptors were further investigated for 168 PFC and non-PFC compounds (Figure S3). The chemical descriptors were particularly computed by the DFT method. For instance, the formation energy, which is defined as the energy required for the formation of a compound from its elemental constituents, was predicted by the DFT-calculated energies of the compound and its elemental constituents. The DFT-calculated frontier orbital analyses were employed for HOMO and LUMO values, while The DFT computation of the dipole-based electric field was used for polarizability. The DFT-calculated energy change associated with oxidation was utilized for ionization energy. Electronic spatial extent gap is a measure of how widely the electronic density is distributed within a molecule. it reveals the average distance between the electrons and the nucleus in a molecule, providing insight into its electronic structure and properties. Electronic spatial extent gap of a molecule was computed by , where  represents the electron wave function of the molecule,  is the distance between an electron and its molecular center, and  is a 3D element of space, meaning the integration of  and  over three-dimensional space. Direct correlations of these descriptors with experimental dielectric strength were further analyzed for the 168 PFC and non-PFC compounds (Figure S3). As seen in the figures, the correlation degree depends on the introduced descriptor.”

Regarding the question on the meaning of the dotted line in Figures 3b, 4b, and 6b, the dotted line describes a normal distribution of the relative errors. We have revised the figure to remove the dotted line for clarity.

Round 2

Reviewer 2 Report

Response: The reviewer raised an important question on the detailed information of the average relative errors. As expected by the reviewer, the relative error of each compound is defined as the deviation of the DFT-calculated value from its experimental value, which is subsequently normalized by the experimental value. The average relative error is therefore the averaged value for the relative errors of 168 PFC and non-PFC compounds. In that sense, the average relative error is a descriptor to determine how accurately the DFT-based modeling approach can predict the core parameters.

As suggested by the reviewer, we have included detailed information of the average relative error in the manuscript.

Reviewer's comment: I still do not understand how for instance  HOMO and HOMO-LUMO gap, and electronic extent gap errors are evaluated. The eigenvalues of orbitals and the extent of electronc density are purely computational characteristics related to the employed DFT model. They are not available experimentally. So, how are you able to calculate the difference between computational and experimental values?

,

Response: Thank the reviewer for the question on the definition of core factors, namely formation energy and electronic spatial extent gap, and their approaches.

The formation energy, which is defined as the energy required for the formation of a compound from its elemental constituents, was predicted by the DFT-calculated energies of the compound and its elemental constituents. In other words, the formation energy is simply the difference in the DFT-calculated energy between the compound and its elemental constituents.

Reviewer's comment: Such a definition implies that formation energies are not accessible experimentally. Electronic energy comes from solving the electronic Schroedinger equation and the value obtained is connot be compared with the experimental value since in real molecules, for instance, zero-point energy is present. Moreover, experimentally one measures the heat of formation rather than the energy of formation. Standard temperature and p are usually 298 K and 1 atm, respectively. Thus one should correct electronic energy for zero-point energy, thermal enthalpy, and pV terms. Only this way, it is possible to convert computational electronic energy to the respective enthalpy and then to the heat of formation. Moreover, under standard conditions the most stable form of e.g. carbon is graphite. How did you calculate the energy of graphite? The chemical equation defining the formation of CH4, for instance, is:

     C(s) + 2H2(g) à CH4(g), where indexes (s) and (g) stand for solid and gaseous phases, respectively. Therefore, to calculate the heat of formation of CH4(g) one needs the enthalpy of C(s) which corresponds to the graphite at standard conditions.

Thus, how do you calculate the difference between the computational and experimental energies of formation if the latter are not available?

Response: Thus, it is not within the scope of this study to explore a rational design direction for the development of novel compounds with high dielectric strengths. Nonetheless, to provide guidance on design direction, we have included information on compounds that exhibit relatively high dielectric strengths within the 168 PFC/non-PFC compounds analyzed

Reviewer's comment: I have a similar impression. Nevertheless, in the conclusion section, the authors write: „The findings of this study will provide useful guidance for the design a suitable protocol 350 to predict the insulating properties of PFC and non-PFC compounds with the aim of rap-351 idly identifying promising insulating gas candidates that can replace SF6.”

In light of the above sentence, my critique is still valid. Their models are able to predict the insulating properties of studied molecules but cannot say if the molecule is able to replace SF6. Modify the indicated sentence accordingly.

Author Response

Reviewer 2

I still do not understand how for instance HOMO and HOMO-LUMO gap, and electronic extent gap errors are evaluated. The eigenvalues of orbitals and the extent of electronc density are purely computational characteristics related to the employed DFT model. They are not available experimentally. So, how are you able to calculate the difference between computational and experimental values?

Response: Please note that HOMO, HOMO-LUMO gap, and electronic extent gap are purely computational quantities. Therefore, we did not evaluate the difference between computational and experimental values for the three quantities in this study. We only correlated the computational quantities with experimental dielectric strength for each organic compound, as shown in Figure S3.

Such a definition implies that formation energies are not accessible experimentally. Electronic energy comes from solving the electronic Schroedinger equation and the value obtained is connot be compared with the experimental value since in real molecules, for instance, zero-point energy is present. Moreover, experimentally one measures the heat of formation rather than the energy of formation. Standard temperature and p are usually 298 K and 1 atm, respectively. Thus one should correct electronic energy for zero-point energy, thermal enthalpy, and pV terms. Only this way, it is possible to convert computational electronic energy to the respective enthalpy and then to the heat of formation. Moreover, under standard conditions the most stable form of e.g. carbon is graphite. How did you calculate the energy of graphite? The chemical equation defining the formation of CH4, for instance, is: C(s) + 2H2(g) à CH4(g), where indexes (s) and (g) stand for solid and gaseous phases, respectively. Therefore, to calculate the heat of formation of CH4(g) one needs the enthalpy of C(s) which corresponds to the graphite at standard conditions. Thus, how do you calculate the difference between the computational and experimental energies of formation if the latter are not available?

Response: As responded in the previous comment, we did not evaluate the difference between computational and experimental values for the formation energy. We only correlated the computational quantity with experimental dielectric strength for each organic compound, as shown in Figure S3.

I have a similar impression. Nevertheless, in the conclusion section, the authors write: „The findings of this study will provide useful guidance for the design a suitable protocol 350 to predict the insulating properties of PFC and non-PFC compounds with the aim of rap-351 idly identifying promising insulating gas candidates that can replace SF6.” In light of the above sentence, my critique is still valid. Their models are able to predict the insulating properties of studied molecules but cannot say if the molecule is able to replace SF6. Modify the indicated sentence accordingly.

Response: As suggested by the reviewer, the last sentence of the manuscript has been revised as follows.

“The findings of this study will provide useful guidance for the design a suitable protocol to predict the insulating properties of PFC and non-PFC compounds.”
